# Remediation of Polycyclic Aromatic Hydrocarbon-Contaminated Soil by Using Activated Persulfate with Carbonylated Activated Carbon Supported Nanoscale Zero-Valent Iron

Changzhao Chen [1], Zhe Yuan [2], Shenshen Sun [2], Jiacai Xie [3], Kunfeng Zhang [3], Yuanzheng Zhai [4], Rui Zuo [4], Erping Bi [1], Yufang Tao [2],* and Quanwei Song [3],*

[1] MOE Key Laboratory of Groundwater Circulation and Environmental Evolution, China University of Geosciences, Beijing 100083, China; chenchangzhao@cnpc.com.cn (C.C.); bi@cugb.edu.cn (E.B.)

[2] College of Chemistry & Environmental Engineering, Yangtze University, Jingzhou 434023, China; 201702666@yangtzeu.edu.cn (Z.Y.); 2023710259@yangtzeu.edu.cn (S.S.)

[3] State Key Laboratory of Petroleum Pollution Control, CNPC Research Institute of Safety and Environment Technology, Beijing 102206, China; xiejc@petrochina.com.cn (J.X.); zhangfweng@163.com (K.Z.)

[4] Engineering Research Center of Groundwater Pollution Control and Remediation of Ministry of Education of China, College of Water Sciences, Beijing Normal University, Beijing 100875, China; zyz@bnu.edu.cz (Y.Z.); zr@bnu.edu.cn (R.Z.)

* Correspondence: author: yftao05@163.com (Y.T.); songquanwei@cnpc.com.cn (Q.S.)

**Abstract:** Soil contamination by polycyclic aromatic hydrocarbons (PAHs) has been an environmental issue worldwide, which aggravates the ecological risks faced by animals, plants, and humans. In this work, the composites of nanoscale zero-valent iron supported on carbonylated activated carbon (nZVI-CAC) were prepared and applied to activate persulfate (PS) for the degradation of PAHs in contaminated soil. The prepared nZVI-CAC catalyst was characterized by scanning electron microscopy (SEM), X-ray diffractometer (XRD), Fourier transform infrared spectroscopy (FTIR), and X-ray photoelectron spectroscopy (XPS). It was found that the PS/nZVI-CAC system was superior for phenanthrene (PHE) oxidation than other processes using different oxidants (PS/nZVI-CAC > PMS/nZVI-CAC > $H_2O_2$/nZVI-CAC) and it was also efficient for the degradation of other six PAHs with different structures and molar weights. Under optimal conditions, the lowest and highest degradation efficiencies for the selected PAHs were 60.8% and 90.7%, respectively. Active $SO_4^{-\bullet}$ and $HO^{\bullet}$ were found to be generated on the surface of the catalysts, and $SO_4^{-\bullet}$ was dominant for PHE oxidation through quenching experiments. The results demonstrated that the heterogeneous process using activated PS with nZVI-CAC was effective for PAH degradation, which could provide a theoretical basis for the remediation of PAH-polluted soil.

**Keywords:** polycyclic aromatic hydrocarbons; soil remediation; nanoscale zero-valent iron; carbonylated activated carbon; persulfate; advanced oxidation process



## 1. Introduction

Polycyclic aromatic hydrocarbons (PAHs) are classic hydrophobic organic compounds (HOCs) consisting of two or more benzene rings linked in the form of condensed rings [1]. PAHs are mainly derived from natural phenomena or human production activities, including natural disasters (such as volcanic eruptions and forest fires), transportation, industrial emissions, and the incomplete combustion of various fossil fuels and other hydrocarbons [2]. Owing to the characteristic of high hydrophobicity, PAHs tend to combine with soil organic matter and clay minerals [3]. PAHs in the soil can be ingested by plants and eventually pose a threat to human health through the food chain due to their bio-accumulation and toxicity [4]. According to the "National Soil Pollution Survey Bulletin", the over-standard rate of contaminated soil by PAHs was 1.4%, and PAHs were mainly detected in farmlands,

industrial zones and wastelands, oil fields, and mining fields [5]. Therefore, the remediation of PAH-polluted soil has aroused widespread concern.

A variety of remediation technologies have been developed, including physical remediation (e.g., steam extraction and thermal desorption) [6,7], chemical remediation (e.g., in situ chemical oxidation and electrokinetic remediation) [1,8], biological remediation (e.g., microbial remediation and phytoremediation) [9], and physicochemical remediation (e.g., solvent extraction/soil washing or soil flushing) [10]. Among these technologies, the physical restoration method has a high energy consumption and may destroy the original structure of the soil [6,11]. The biological methods hold the drawbacks of a long restoration period and the feasibility of biological methods largely depends on the limiting factors and the location of pollutants [11,12]. In contrast, chemical oxidation, as a promising remediation technology, converts organic pollutants into harmless or less harmful chemicals after the introduction of chemical oxidants directly into pollution sources, which has advantages in cost and efficiency [13–15]. The commonly used chemical oxidants include ozone ($O_3$), hydrogen peroxide ($H_2O_2$), permanganate ($KMnO_4$), and persulfate (PS). However, the poor stability of $O_3$ and $H_2O_2$, and the high affinity with soil organic matter of $KMnO_4$, make these oxidants difficult to transport to designated areas [16]. On the contrary, PS is advantageous in the remediation of organic contaminated soil due to its high redox potential ($S_2O_8^{2-}$, $E^0 = 2.1$ V) and chemical stability [1,16]. Active free radicals (e.g., $SO_4^{-\bullet}$ and $HO^{\bullet}$) and non-free radicals ($^1O_2$) are generated by the breakage of the O-O bond in PS molecules, thus rapidly mineralizing various persistent organic pollutants [17]. Moreover, PS used in the remediation process finally decomposes to the main derivative of $SO_4^{2-}$, and $SO_4^{2-}$ can combine with other soil substances to generate sulfate, which is one of the inherent components of soil [18–20]. Therefore, PS has been widely applied in the processes of in situ chemical oxidation for the remediation of various forms of organics-contaminated soil [21–23]. Compared to other oxidants, PS has a lower price [24], so PS has a higher economic value.

PS can be activated by heat [25], alkali [26], transition metals [14], ultrasound (US) [25], and ultraviolet (UV) irradiation [10]. Among them, heterogeneous catalysts, such as natural metallic minerals, carbon materials, and transition metal oxides, exhibit excellent performance in PS activation [1,17,27].

As an environmentally friendly and cheap metal element, iron is the first choice for the activation of persulfate [28]. Nano zero-valent iron (nZVI) has a high specific surface area, surface energy, reactivity, and reduction properties, and the direct or indirect generation of $Fe^{2+}$ can effectively activate PS [17]. However, it has been reported that it is easy for a single nZVI to agglomerate and be affected by acid and alkali, and the formation of hydroxide after the reaction leads to a large area of passivation, undermining its activity [3,29]. For example, it was reported that the agglomeration of conventional nZVI particles affected the activation of PS and consequently inhibited the remediation of anthracene in soil [9]. To overcome the limitations, composite materials composed of nZVI particles loaded onto stable materials, such as activated carbon (AC), biochar (BC), and zeolite, were developed [30–32]. Among these supporting materials, AC was favored in the reported research due to its low cost, high specific surface area, richness of porous structure, and surface functional groups [33,34]. For example, Zhang et al. reported the activation of PS by synthesized AC-nZVI for ampicillin oxidation, and it was found that the loaded catalyst effectively inhibited nZVI particle agglomeration and Fe ion leakage [33]. Among many activated carbon preparation materials, coconut shell is one of the ideal materials because of its abundant supply in many countries. Moreover, the natural structure of coconut shell is suitable for the preparation of porous materials with low ash content [35]. In addition, several studies have reported that the AC could be utilized as a catalyst to activate PS for the decomposition of soil organic pollutants [1,36]. The oxygen-containing carbonyl functional group (C=O) and $sp^2$ hybridized carbon ($sp^2$-C) of AC itself could act as active sites for the activation of PS [16,37]. It has been reported that PS was specifically adsorbed to the active sites of the porous structure of AC, and subsequently, $^1O_2$ was

generated by these active sites (e.g., C=O and sp$^2$-C) reacting with PS to degrade organic pollutants via non-radical pathways [38]. Cheng and co-authors found that, among various oxygen-containing functional groups of carbon nanotubes, the C=O group was the main active site that promoted PS activation for 2,4-dichlorophenol oxidation [39]. Therefore, the development of composites of carbonylated activated carbon (CAC) with nZVI seems to be a feasible strategy for PS activation to enhance the oxidation of soil pollutants. In this scenario, the modified AC with rich C=O groups not only functioned as a supporting material for stabilizing nZVI but concurrently played a role in the synergistic activation of PS. Nevertheless, to the knowledge of the authors, the synthesis of CAC-supported nanoscale zero-valent iron to be applied in the ISCO process for the remediation of PAH-contaminated soil has not been reported so far.

Herein, composites of nanoscale zero-valent iron supported on carbonylated activated carbon (nZVI-CAC) were prepared by the chemical reduction method for the activation of PS to remove PAHs from soil. PHE is one of the 16 polycyclic aromatic hydrocarbons listed as priority pollutants [40] which is biotoxic, teratogenic, mutagenic, and difficult to degrade [41,42]. In addition, PHE is reported to be ubiquitous in the environment, especially in contaminated soils [43–46]. Therefore, PHE is selected as the representative of PAHs.

The objectives of this study are as follows: (1) to explore the catalytic performance of nZVI-CAC for soil pollutant oxidation using PHE; (2) to optimize the operation conditions and to evaluate the feasibility of oxidation of different PAHs in soil; and (3) to elucidate the intrinsic mechanism of the developed PS/nZVI-CAC system.

## 2. Results and Discussions

### 2.1. Characterization of CAC and nZVI-CAC

The XRD patterns of CAC and nZVI-CAC are shown in Figure 1a. The peaks of CAC at 2θ of 26.5° and 44.3° were attributed to the high graphitization of the carbon material [47]. The peak of nZVI-CAC at 44.6° represented a typical reflection of Fe$^0$, indicating a successful loading of Fe [48].

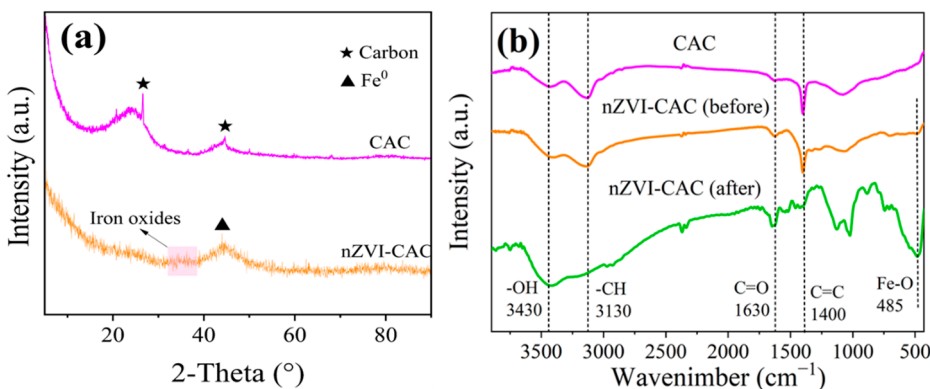

**Figure 1.** (**a**) XRD pattern of CAC and nZVI-CAC; (**b**) FTIR spectra of CAC and nZVI-CAC before and after the reaction.

The FTIR spectra of CAC and nZVI-CAC reflected the types and changes of groups on the sample surface (Figure 1b). The distinct absorption peaks at 3430 cm$^{-1}$, 3130 cm$^{-1}$, and 1630 cm$^{-1}$ for both samples represented the -OH stretching vibration, -CH$_2$ asymmetric stretching vibration, and C=O stretching vibration, respectively [49,50]. The peak at 1400 cm$^{-1}$ could be attributed to the stretching vibrations of C=C on the surface of the samples [32]. The absorption peaks at 620–400 cm$^{-1}$ were from the stretching vibrations of Fe-O bonds of different iron oxides, but the non-sharp absorption peaks indicated that Fe was mainly presented in the form of Fe$^0$ with a low content of Fe oxides [32]. After the reaction, the O-H stretching vibration at 3430 cm$^{-1}$ was enhanced, indicating that PHE was adsorbed to nZVI-CAC [51]; the enhancement of the Fe-O peak can be attributed to the

oxidation of Fe$^0$ [52], the enhancement of C=O may be due to the transfer of electrons from PHE to nZVI-CAC [53].

The SEM images of CAC and nZVI-CAC are revealed in Figure 2a–c. The CAC surface was rich in pore structures, which could provide a large number of loading sites for active iron species. As shown in Figure 2c, many fine particles were attached to the nZVI-CAC surface, and the average particle size was close to 100 nm, further indicating that the Fe nanoparticles were successfully loaded onto the surface of CAC. The black spots in the TEM characterization of nZVI-CAC (Figure 2d–h) are nZVI particles on the surface of CAC. nZVI presents a chain structure, which is caused by magnetic interaction between particles and surface tension [54]. In addition, the lattice fringe spacing of nZVI-CAC was determined to be 0.202 nm, which corresponded to the (110) plane of Fe$^0$ [55]. It can be seen from the EDS mapping of nZVI-CAC that there are abundant Fe elements on the surface of CAC. The TEM and EDS characterization of nZVI-CAC further verified the successful loading of Fe$^0$ onto CAC.

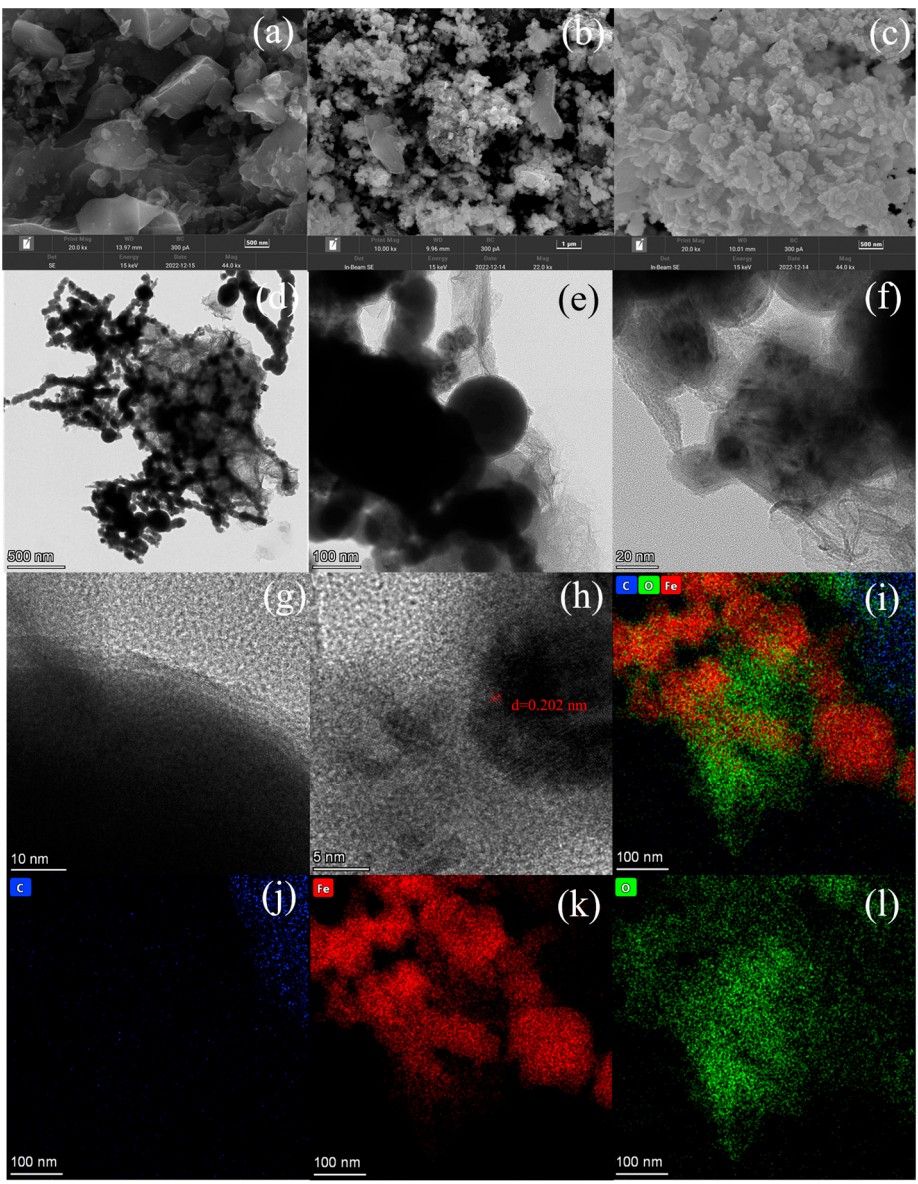

**Figure 2.** (**a**) SEM of CAC; (**b**,**c**) SEM of nZVI-CAC; (**d–h**) TEM of nZVI-CAC; (**i–l**) EDS of nZVI-CAC.

The BET-specific surface areas of CAC and nZVI-CAC were 412.8 $m^2$ $g^{-1}$ and 147.8 $m^2$ $g^{-1}$, respectively. The large surface area of CAC was attributed to the abundance of pore structures. However, the specific surface area of nZVI-CAC formed after $Fe^0$ loading decreased, which might be due to the blockage of pores caused by nZVI particles covering the pores of CAC [48]. As shown in Figure 3a,b, the $N_2$ isotherms of CAC displayed a typical mesoporous carbon material (IUPAC, type IV). The main $N_2$ adsorption in the lower pressure range ($P/P_0 < 0.1$) also indicated the existence of microporous structures [56], and the average pore size was 2.02 nm. The nZVI-CAC exhibited a type IV adsorption isotherm with an H3 hysteresis loop ($0.4 < P/P_0 < 1.0$) and an average pore size of 3.80 nm, implying that mesopores were mainly present in the catalyst after nZVI loading.

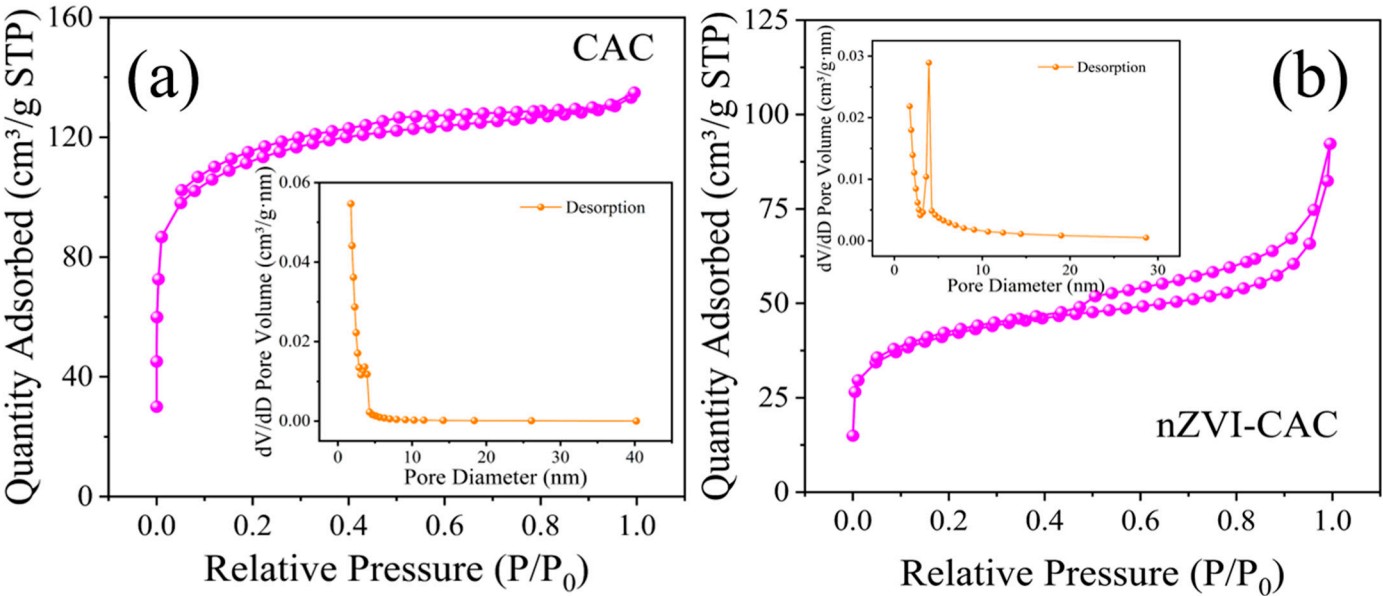

**Figure 3.** The adsorption–desorption isotherms and the pore size distribution (the insets of figure) of CAC (**a**) and nZVI-CAC (**b**).

The characteristic peaks of C 1s, O 1s, and Fe 2p were observed in the XPS full-scale spectrum of CAC and nZVI-CAC (Figure 4a). As illustrated in Figure 4b, the peaks at 284.8 eV, 286.4 eV, 288.5 eV, and 290.7 eV in the C1s spectra of CAC and nZVI-CAC were convolved as $sp^2$-C, $sp^3$-C, C-O, and C=O, respectively [57]. The relative content of $sp^3$-C increased from 19.7% to 26.9% after loading Fe, which indicates that nZVI-CAC was produced [57,58]. The characteristic peaks of the O 1s spectra were C=O (531.2 eV), C-OH or Fe-O (532.2 eV), C-C=O (533.3 eV), and C-O-C (534.5 eV), respectively [59]. It can be seen from Figure 4c that the relative C-OH/Fe-OH concentration of nZVI-CAC increased significantly, suggesting that the Fe species loaded onto the surface of CAC mainly exist in the form of Fe oxides [29,30]. The Fe 2p spectra of nZVI-CAC are shown in Figure 4d. The obvious peaks of $Fe^0$ emphasized again the successful loading of nZVI particles, while the marked peaks of $\equiv Fe^{2+}$ and $\equiv Fe^{3+}$ were attributed to the incomplete reduction and the inevitable oxidation of $Fe^{2+}$ during the preparation period [48].

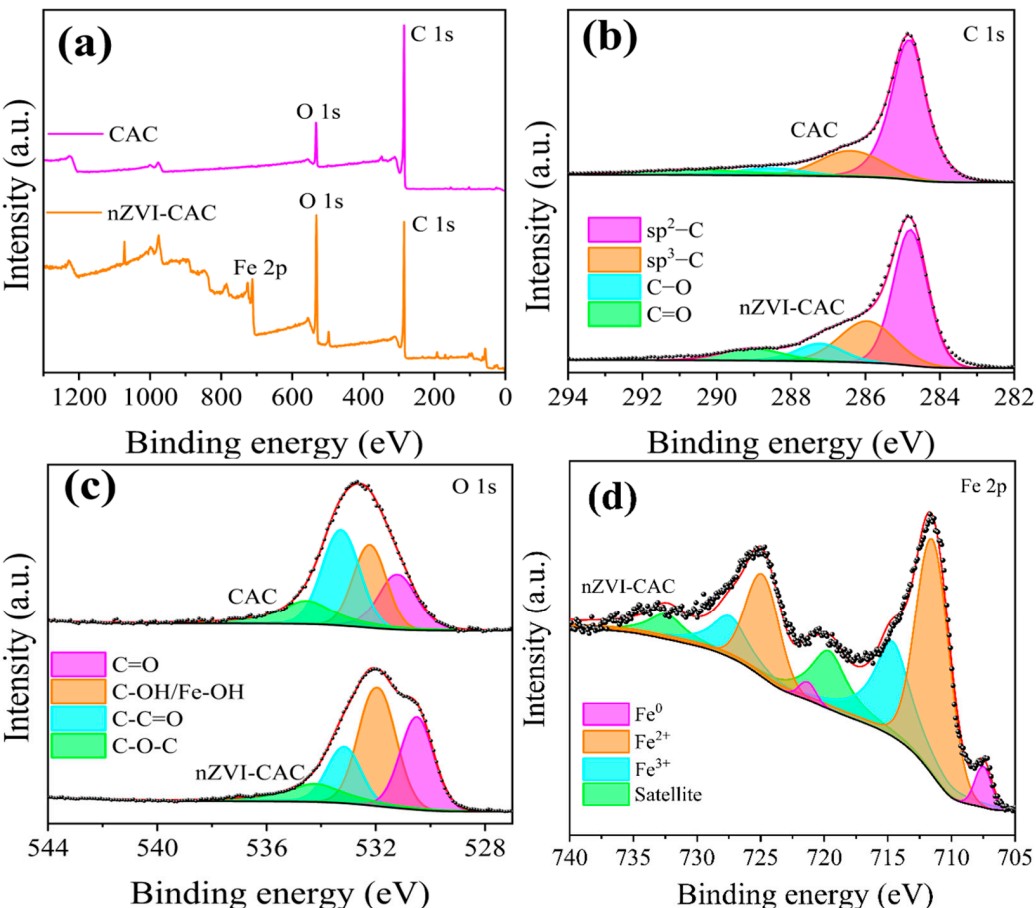

**Figure 4.** (**a**) The full-scale XPS spectrum; (**b**) the C 1s XPS analysis; (**c**) the O 1s XPS analysis of CAC and nZVI-CAC; (**d**) the Fe 2p analysis of nZVI-CAC.

## 2.2. The Greenness of the Analysis Method

The AGREE software (Analytical Greenness Calculator v. 0.5 beta) is used to evaluate the greenness of the analysis method [60]. Using the open-source AGREE software, we can obtain a circular pictograph with 12 numbers, each scale is associated with a color scale from dark green to dark red, the width of each segment represents its weight, and the middle number in the pictograph represents the final average value calculated from the 12 data points [61]. The comprehensive value of this experimental analysis method is 0.61 (Figure 5), which confirms the green character of this experimental analysis method.

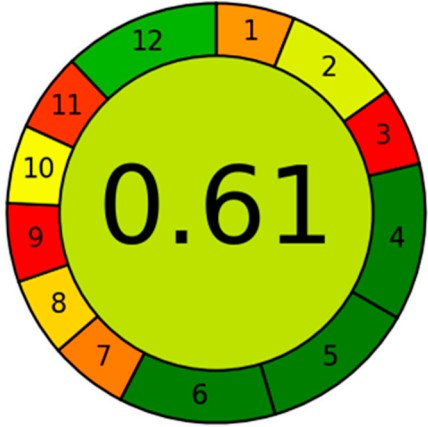

**Figure 5.** The greenness of the analysis method.

### 2.3. Remediation of Soil PHE by Different Systems

The remediation efficiencies of PHE in soil by PS, nZVI-CAC, PS/nZVI, PS/CAC, and PS/nZVI-CAC are compared in Figure 6a. The efficiency of PHE oxidation by PS alone was only 7.4%, which might be due to the decomposition of PS by soil organic matter (SOM) and Fe/Mn-minerals [62]. A removal efficiency of 11.4% for PHE was found by nZVI-CAC alone, revealing the adsorption capacity for soil PHE by nZVI-CAC. The adsorption sites of nZVI-CAC might be provided by the micropores and the abundant oxygen-containing functional groups [48]. The radiation efficiency of the PS/nZVI system for PHE in soil was 16.8%. The limited oxidation efficacy of PHE by the PS/nZVI system could be explained by the following reasons. Owing to the buffering properties of the soil matrix, it was difficult for the soil and water mixtures to maintain low pH values. In this scenario, the superficial $Fe^0$ particles were oxidized to produce $\equiv Fe^{3+}$ and $\equiv Fe^{2+}$ hydroxides and precipitated on the surface of agglomerates, hindering the exchange of $Fe^0$ inside the agglomerates with external ions [28]. Consequently, the sustainable release of $\equiv Fe^{2+}$ and activity of $Fe^0$ was blocked (Equation (1)), and the yielded reactive species were limited through PS activation (Equations (2) and (3)). It is interesting to observe that the removal efficiency of PHE by the PS/CAC system reached 33.2%. The free radical pathway of PS/CAC is that CAC can provide active sites to promote electron transfer, and free radicals are formed near or on the surface of the CAC [63,64]. For example, the abundant oxygen-containing functional groups, such as carbonyl groups on the surface of CAC, could catalyze the decomposition of PS to produce $SO_4^{-\bullet}$ and $HO^\bullet$ for the attack of organic pollutants through the free radical pathway [65]. In addition, carbon materials were reported to be the transmission medium of electrons to induce PS activation by electron transfer, promoting the oxidation of organic pollutants [16,66] and the surface active functional groups, such as C=O on CAC, to activate PS to produce $^1O_2$ [63,67]. Therefore, the PHE was oxidized through the radical and/or nonradical pathways in the PS/CAC process. However, the active sites on the surface of CAC were, after all, restrained, and strategies for further prosperity were necessary. Notably, the removal of PHE in the PS/nZVI-CAC system (60.3%) was significantly improved in comparison with the PS/CAC process, indicating the superiority of the composite catalysts. The corresponding first-order rate constant of PHE degradation in the PS/nZVI-CAC system is 0.00179 $min^{-1}$ (Figure 6b). The supported materials could overcome the deficiency of commercial nZVI, which was prone to passivation and agglomeration, and, meanwhile, compensate for the deficient active sites of CAC.

As shown in Figure 6c, the performance of PS was further compared with $H_2O_2$ and PMS, the other two commonly used oxidants in advanced oxidation techniques for soil remediation [28]. All three oxidants alone performed poorly in terms of PHE removal, and the removal rates by the three oxidants were less than 10%. Only relying on the redox potential to oxidize organic pollutants, the introduction of oxidants alone in soil had been reported to make it difficult for the oxidants to be broken down and for contaminants to be removed effectively [65]. With the addition of nZVI-CAC, the removal efficiencies of PHE followed the order of PS/nZVI-CAC (60.3%) > PMS/nZVI-CAC (54.7%) > $H_2O_2$/nZVI-CAC (51.3%), respectively. All three oxidants could be activated to produce reactive oxygen species (ROS) to attack PHE in soil (Equations (2)−(5)); however, it was reported that the presence of $H_2O_2$ aggravated the hydroxide covering the surface of nZVI-CAC and reduced the available $\equiv Fe^{2+}$ [65]. In contrast, both PMS and PS contained peroxy bonds, which were broken to generate ROS, but the activation of PMS was more difficult resulting in a slightly lower removal efficiency of PHE in the PMS process than in the PS system [68]. Notably, the removal of PHE in the PS/nZVI-CAC system (60.3%) was significantly improved in comparison with the PS/CAC process, indicating the superiority of the composite catalysts. The corresponding first-order rate constant of PHE degradation in the system of PS/nZVI-CAC was 0.00179 $min^{-1}$ (Figure 6b). The supported materials could overcome the deficiency of commercial nZVI, which was prone to passivation and agglomeration, and,

meanwhile, compensate for the deficient active sites of CAC. Therefore, PS was selected as the oxidant, considering efficacy, economic cost, and environmental impact.

$$Fe^0 + 2H_2O \rightarrow Fe^{2+} + 2OH^- + H_2 \tag{1}$$

$$Fe^0 + 2S_2O_8^{2-} \rightarrow Fe^{2+} + 2SO_4^{2-} + 2SO_4^{-\bullet} \tag{2}$$

$$\equiv Fe^{2+} + S_2O_8^{2-} \rightarrow \ \equiv Fe^{3+} + SO_4^{-\bullet} + \ SO_4^{2-} \tag{3}$$

$$\equiv Fe^{2+} + HSO_5^- \rightarrow \ \equiv Fe^{3+} + SO_4^{-\bullet} + OH^- \tag{4}$$

$$\equiv Fe^{2+} + H_2O_2 \rightarrow \ \equiv Fe^{3+} + HO^\bullet + OH^- \tag{5}$$

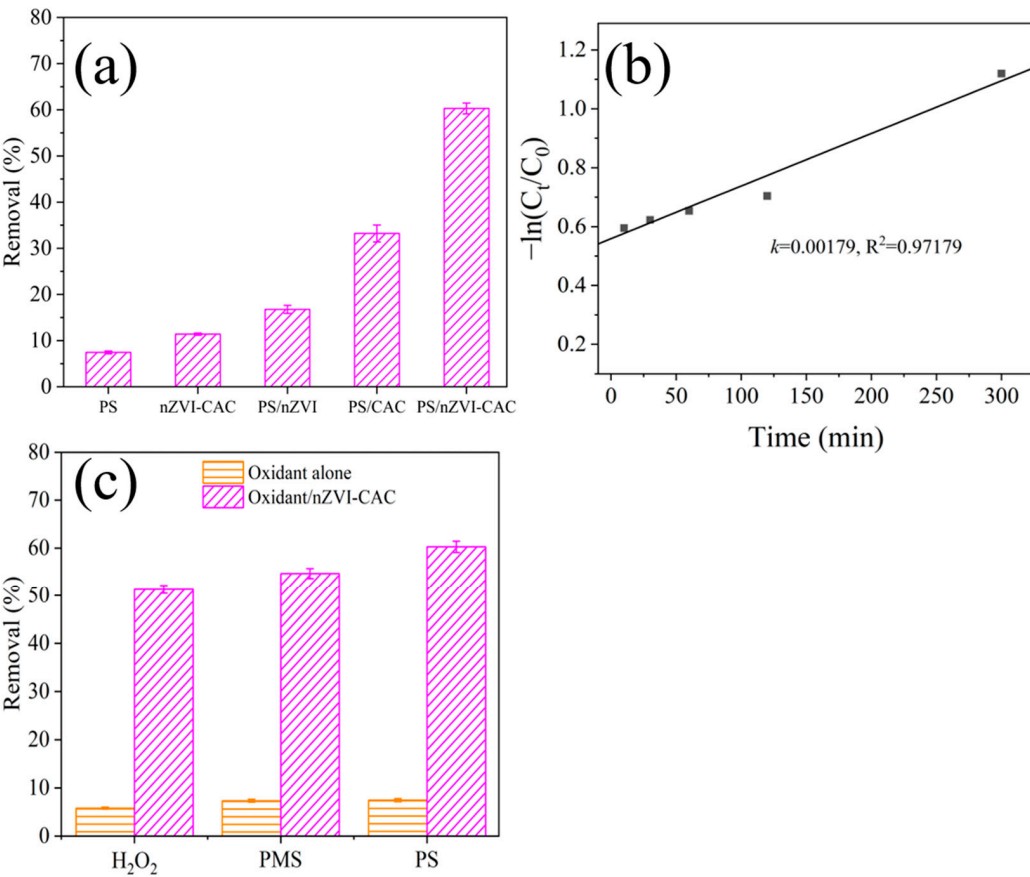

**Figure 6.** (**a**) PHE oxidation in different systems; (**b**) pseudo first order kinetic fit under optimal conditions; (**c**) PHE oxidation by using different oxidants. Conditions: except for the tested factor, the others were fixed, oxidant = 100 mmol kg$^{-1}$, [Catalyst] = 10 g kg$^{-1}$.

### 2.4. Effects of Parameters

The parameters of nZVI-CAC dosage, Fe/CAC mass ratio, and oxidant concentration greatly affected the remediation process, and these factors need to be optimized for the development of an optimal remediation system. As recorded in Figure 7a, the remediation of PHE with different Fe/CAC mass ratios was studied. With the Fe/C mass ratio increasing from 0.4:1 to 4:1, the removals of soil PHE under nZVI-CAC alone were in the range of 6.4–13.7%. In the PS-involved systems, the removals of PHE exhibited an augment trend, with the Fe/CAC mass ratio increasing from 0.4:1 to 1:1, while the PHE removal efficiencies subsequently decreased as the mass ratio of Fe/CAC further increased to 4:1. With the Fe/CAC mass ratio increasing from 0.4:1 to 1:1, the loaded active iron was amplified and more active sites could be provided by the composites. However, with the occurrence of Fe overloading, the Fe$^0$ particles appeared to agglomerate and fill up the pore channels, resulting in the possible shrinkage of the specific surface areas of the catalysts and also the

restriction of PS activation by the active sites in the internal region of the catalysts [48,69]. Therefore, the Fe/CAC mass ratio of 1:1 was favored for the oxidation of contaminants.

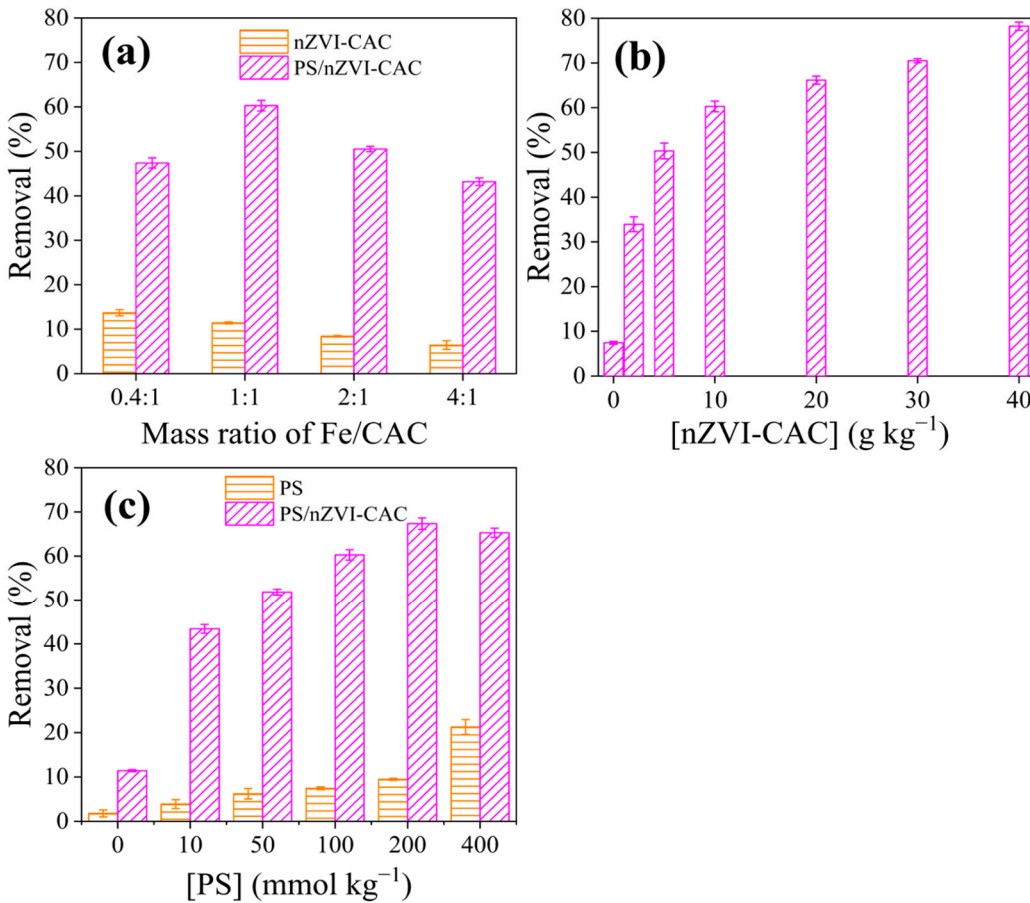

**Figure 7.** (**a**) The effects of mass ratio of Fe/CAC; (**b**) the nZVI-CAC dosage on PHE oxidation; (**c**) PS concentration on PHE oxidation. Except for the tested factor, the others were fixed, Fe/CAC = 1:1, [PS] = 100 mmol kg$^{-1}$, [Catalyst] = 10 g kg$^{-1}$.

As shown in Figure 7b, as the dose of nZVI-CAC was increased from 0 to 10 g kg$^{-1}$, the removal efficiency of soil PHE increased rapidly from 7.4% to 60.3%, while, upon further addition of nZVI-CAC to 40 g kg$^{-1}$, the removal rate tardily increased to 78.2%. The positive correlation of the PHE removal efficiency with the dose of nZVI-CAC possibly could be ascribed to the increased amount of active sites provided by the augment of nZVI-CAC for both the adsorption and activation of PS [37,70]. The sluggish elevation of PHE removal was probably due to the limited concentration of PS [17]. The dosage of nZVI-CAC was selected as 10 g kg$^{-1}$, taking the cost of catalysts into consideration.

The effect of PS dosage on PHE removal was also considered and investigated (Figure 7c). When nZVI-CAC was not introduced, the oxidation efficiency of PHE slightly increased with an increasing PS dosage, revealing the possibility of limited PHE removal by a combination of PS relying on its own oxidizing properties and ROS generation by soil Fe/Mn minerals and SOM. On the contrary, in the PS/nZVI-CAC systems with various PS doses, the removal efficiencies of PHE did not increase consistently with the increase in PS concentration, and the maximum removal efficiency was reached at a PS concentration of 200 mmol kg$^{-1}$. The inhibition of PHE oxidation at high doses of PS might be attributed to the mutual quenching of radicals generated by excess PS, which led to a lower content of ROS in the system and the inhibition of the oxidation capacity of soil PHE [17,71].

## 2.5. Oxidation Performance for Various PAHs

The toxicity of PAHs varies with the number of rings. Generally, PAHs with five rings and above were less hydrophilic, and their strong interaction with soil organic matter causes them to be confined and difficult to utilize by organisms [72]. In contrast, PAHs with 3–4 rings were more likely to enter soil pore water, resulting in high bioaccessibility and great ecological risk [73]. Therefore, six additional 3–4 ring-PAHs (Table 1) were selected to investigate the oxidation feasibility by the PS/nZVI-CAC system. As depicted in Figure 8, the removal efficiencies of the seven tested PAHs were 90.7% for ACP, 86.5% for ACE, 72.6% for FLU, 67.4% for PHE, 67.4% for ANT, 62.2% for FLUA, and 60.8% for PYR. It was observed that the removal of different PAHs was reduced with an increase in molecular weight (Table 1). Owing to the poor water solubility of high molecular weight PAHs, the removal of PAHs with high molecular weights was more difficult [74]. With PS alone, the correlation between the removal and the molecular weights of PAHs was not observable due to the relatively low amounts of removal of PAHs by PS.

**Table 1.** Abbreviations and structural formulas of the tested PAHs.

| Reagents | Abbreviation | Molecular Weight | Structural Formula |
|---|---|---|---|
| Acenaphthene | ACP | 154.21 | |
| Acenaphthylene | ACE | 152.19 | |
| Fluorene | FLU | 166.22 | |
| Phenanthrene | PHE | 178.23 | |
| Anthracene | ANT | 178.23 | |
| Fluoranthene | FLUA | 202.25 | |
| Pyrene | PYR | 202.25 | |

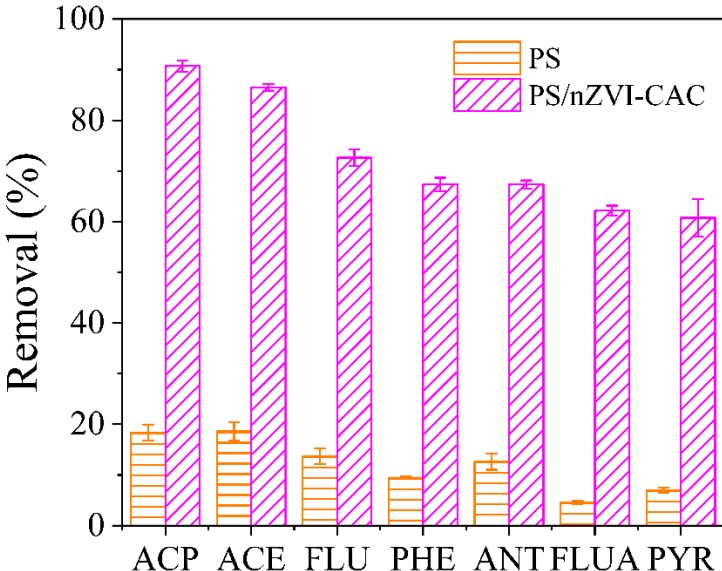

**Figure 8.** Oxidation of various PAHs by the system of PS and PS/nZVI-CAC. Conditions: [PS] = 200 mmol kg$^{-1}$, [Catalyst] = 10 g kg$^{-1}$.

### 2.6. Catalytic Mechanism of the PS/nZVI-CAC Process

During the catalytic processes, ROS including SO$_4^{-\bullet}$, HO$^{\bullet}$, O$_2^{-\bullet}$, $^1$O$_2$ were probably produced and likely contributed to the degradation of PAHs. To clarify the dominant ROS and to elucidate the potential mechanism of PS activation by nZVI-CAC, quenching experiments with 500 mmol kg$^{-1}$ MeOH, TBA, CF, FFA, and phenol were carried out. MeOH was commonly involved in scavenging both HO$^{\bullet}$ ($k_{\text{MeOH, HO}^{\bullet}}$ = 9.7 × 10$^8$ M$^{-1}$ s$^{-1}$) and SO$_4^{-\bullet}$ ($k_{\text{MeOH,SO}_4^{-}\bullet}$ = 2.5 × 10$^7$ M$^{-1}$ s$^{-1}$), and TBA predominantly quenched HO$^{\bullet}$ ($k_{\text{TBA, HO}^{\bullet}}$ = 5.2 × 10$^{10}$ M$^{-1}$ s$^{-1}$) in bulk solution [75]. As illustrated in Figure 9, The addition of MeOH and TBA resulted in the reduction of the PHE removal efficiencies by 24.5% and 19.7%, respectively, indicating that HO$^{\bullet}$ and SO$_4^{-\bullet}$ were the contributive ROS in the PS/nZVI-CAC system. However, it is worth noting that the PS/nZVI-CAC system presented an advantage for PHE removal over the H$_2$O$_2$/nZVI-CAC process (Figure 6b), suggesting SO$_4^{-\bullet}$ was more selective and contributive towards PHE than HO$^{\bullet}$ [76–79]. In addition, the general electropositivity of the composite of nZVI supported by AC or BC, as well as the abundant C=O on the catalysts, favored the adsorptions of PS [70,80,81], which probably led to the production of absorbed SO$_4^{-\bullet}$ via the reactions of PS with Fe$^0$, ≡Fe$^{2+}$ (Equations (2) and (3)), and also the C=O group (Equation (6)) during the catalytic reactions of nZVI-CAC [37]. Therefore, the slightly better inhibitory effect of MeOH than that of TBA probably could be attributed to the overwhelming active SO$_4^{-\bullet}$ mainly presenting on the surface of nZVI-CAC where the absorbed SO$_4^{-\bullet}$ radical was not quenched by MeOH due to the hydrophilic nature of MeOH [82]. O$_2^{-\bullet}$ could be produced by the redox reaction between superficial ≡Fe$^{2+}$ on the solid and dissolved oxygen (Equation (7)) [28], and O$_2^{-\bullet}$ was usually quenched by CF ($k_{\text{CF, O}_2^{-}\bullet}$ = 3.0 × 10$^{10}$ M$^{-1}$ s$^{-1}$) [83]. When CF was present, PHE removal was inhibited by 14.3% in this study. Considering the low reactively of O$_2^{-\bullet}$ itself with pollutants, the inhibition of PHE oxidation by CF could be ascribed to the participation of O$_2^{-\bullet}$ for the activation of PS to produce SO$_4^{-\bullet}$ for PHE degradation (Equation (8)) [84].

$$\text{nZVI} - \text{CAC} - \text{C} = \text{O} + \text{S}_2\text{O}_8^{2-} \rightarrow \text{nZVI} - \text{CAC} - \text{C} - \text{O}^* + 2\text{SO}_4^{-\bullet} \tag{6}$$

$$\equiv \text{Fe}^{2+} + \text{O}_2 \rightarrow \text{O}_2^{-\bullet} + \equiv \text{Fe}^{3+} \tag{7}$$

$$\text{O}_2^{-\bullet} + \text{S}_2\text{O}_8^{2-} \rightarrow \text{SO}_4^{2-} + \text{SO}_4^{-\bullet} + \text{O}_2 \tag{8}$$

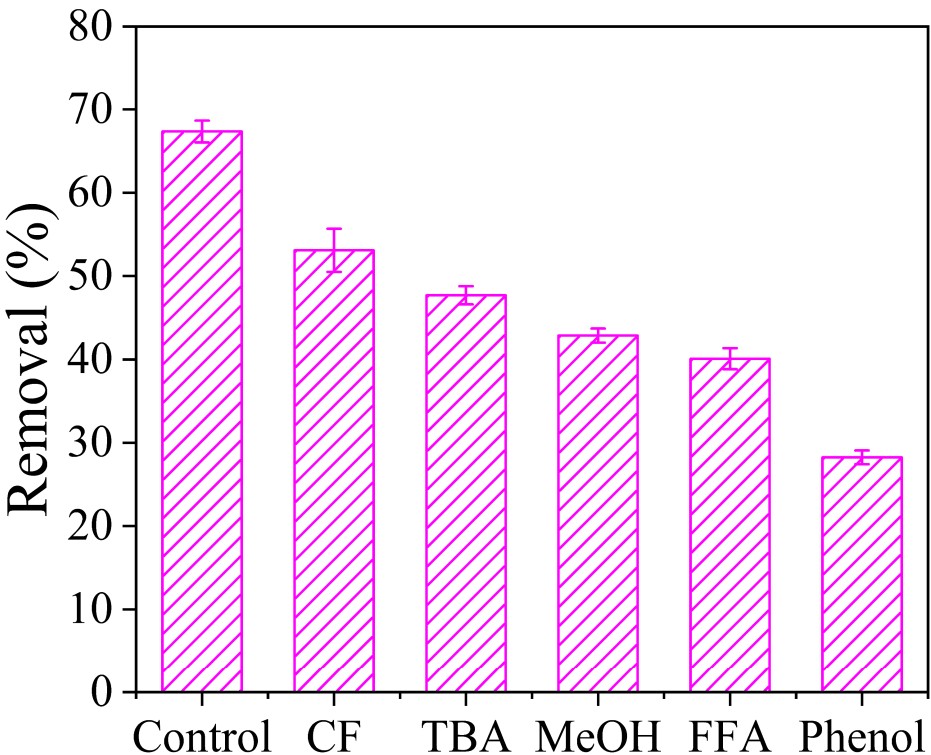

**Figure 9.** The effects of radical scavengers on PHE oxidation; conditions: [PS] = 200 mmol/kg, [nZVI-CAC] = 10 g kg$^{-1}$, [Scavenger] = 500 mmol kg$^{-1}$.

In the presence of FFA, the PHE removal efficiency was observably restrained by 27.3%, which was higher than the inhibition using MeOH. FFA was generally used as an inhibitor of $^1O_2$ ($k_{\text{FFA},1O_2}$ = 1.2 × 10$^8$ M$^{-1}$ s$^{-1}$), and $^1O_2$ was reported to be the dominant ROS produced by activation of PS [85]. However, Lu et al. argued that these studies ignored the rapid quenching of $^1O_2$ in water in the non-optical regime and the false-positive results of electron paramagnetic resonance (EPR) experiments [86]. More importantly, FFA was also capable of quenching the active $SO_4^{-\bullet}$ and $HO^\bullet$ species bound to the surface of the solid catalysts due to the hydrophobicity of FFA [82,87,88]. This led to the higher inhibition of PHE degradation with FFA than with MeOH. To further verify this viewpoint, phenol was introduced as a scavenger since phenol was reported to effectively scavenge $HO^\bullet$ ($k_{\text{phenol, }HO^\bullet}$ = 6.6 × 10$^9$ M$^{-1}$ s$^{-1}$) and $SO_4^{-\bullet}$ ($k_{\text{phenol},SO_4^{-}\bullet}$ = 8.8 × 10$^9$ M$^{-1}$ s$^{-1}$) present in a bound state on the surface of solid materials [82,85,89]. With the addition of phenol, PHE oxidation was dramatically retarded by 39.1%, which was higher than that of FFA. Nevertheless, owing to the much lower rate constant with $^1O_2$ ($k_{\text{phenol, }1O_2}$ = 2.6 × 10$^6$ M$^{-1}$ s$^{-1}$), phenol is generally not regarded as a scavenger of $^1O_2$ [78]. As a consequence, it could be concluded that surface-bound $SO_4^{-\bullet}$ was predominantly responsible for PHE oxidation and $^1O_2$ was not the main ROS contributor to PHE oxidation.

Based on the above discussion, the mechanism of PS activation by nZVI-CAC for the oxidative remediation of organic contaminants is proposed in Figure 10. Specifically, the PS molecules were specifically adsorbed to nZVI-CAC rich in C=O groups, and then $SO_4^{-\bullet}$ and $HO^\bullet$ were generated through reactions between PS and ≡Fe$^{2+}$, Fe$^{0\prime}$ and C=O in the nZVI-CAC material to effectively attack soil PHE. In the meantime, dissolved oxygen could react with ≡Fe$^{2+}$ to produce $O_2^{-\bullet}$, and $O_2^{-\bullet}$ subsequently activated PS to enhance the production of $SO_4^{-\bullet}$ and $HO^\bullet$. During the catalytic process, the presence of C=O groups on nZVI-CAC functioned as both adsorptive sites and active sites for PS, strengthening the formation of surface-bound $SO_4^{-\bullet}$ and $HO^\bullet$ for the remediation of soil pollutants.

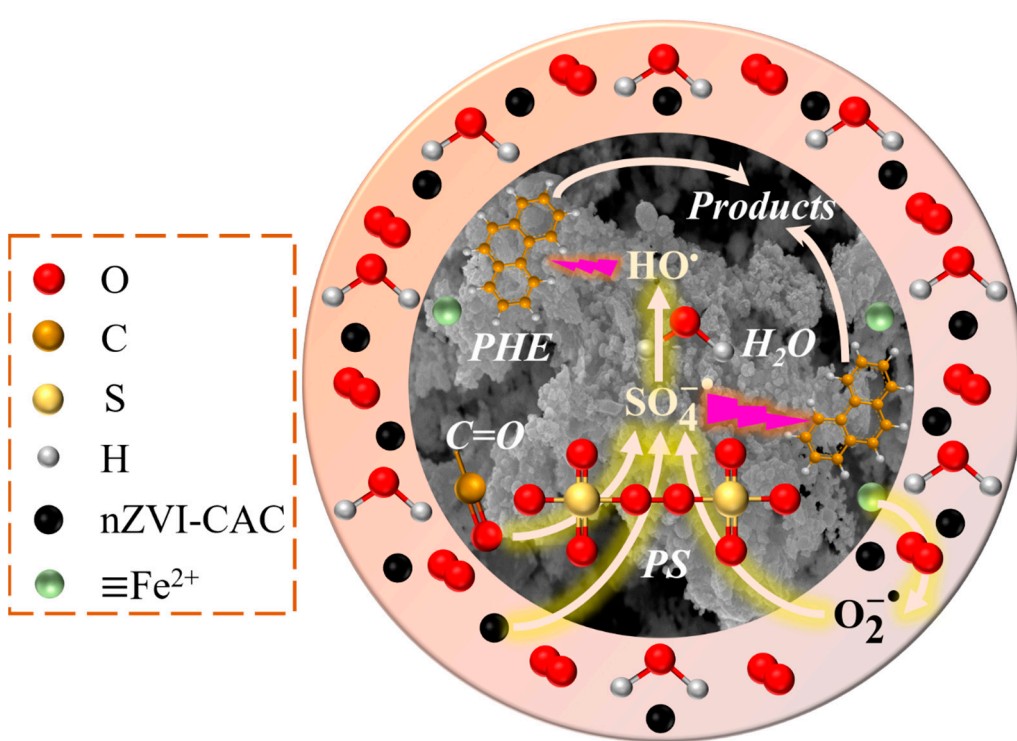

**Figure 10.** Mechanism of PS/nZVI-CAC system for PHE oxidation.

*2.7. Degradation Products of PHE*

The intermediates produced during the degradation of PHE by nZVI-CAC/PS were identified and are listed in Table 2. Three possible degradation pathways for PHE were summarized based on the products. In the first pathway, PHE was first attacked by ROS to produce 9-phenanthrol at the C9,C10 position of the PHE central ring. The 9-phenanthrol was hydroxylated to 9, 10-dihydro-9,10-dihydroxyphenanthrene (D). D was then oxygenated to 2,2′-biphenyldicarboxylic anhydride (B) [90]. B could be isomerized to biphenyldicarboxylic acid (A) by ring opening [91]. A was further oxidized to a short-chain acid. D can also form 2,2-biphenyldicarbalin by oxidation and ring-opening, and 3,4-benzocoumarin (F) can be produced by oxidation and esterification. Additionally, amiable nucleus addition could generate 9,10-phenanthrenequinone, and 9,10-phenanthrenequinone was oxygenated to B [91]. In the second pathway, PHE was attacked by C3 to generate [1,1′-biphenyl]-2-acetic acid (C). C could further decompose to 1,2,4-trimethylbenzene through chain breaking or to 1,2,3-trimethylbenzene through methylation and oxidation [92]. In the third pathway, PHE could be attacked at the C2 and C3 positions forming 2,3-phenanthrenediol. The 2,3-phenanthrenediol further isomerizes to 2,3-phenanthrenequinone [93], followed by ring opening to 1,2-naphthalic anhydride (E) [90]. E was further hydroxylated to 1,4-dihydroxy-2-naphthalic acid through oxidation, ring-opening, and loss of $CO_2$ [94]. The 1,4-dihydroxy-2-naphthalic acid is then decarboxylated to l-(+)-mandelic acid, or hydroxylated to 2,4-dihydroxybenzoic acid (G), and those intermediates are further oxidized to short-chain acids [43].

**Table 2.** PHE degradation products determined by GC-MS.

| Products | Chemical Name | M.W. | Chemical Structure |
|---|---|---|---|
| A | 2,2′-Biphenyldicarboxylic acid | 242 | |
| B | 2,2′-Biphenyldicarboxylic anhydride | 224 | |
| C | [1,1′-Biphenyl]-2-acetic acid | 212 | |
| D | 9,10-Dihydro-9,10-dihydroxyphenanthrene | 212 | |
| E | 1,2-Naphthalic anhydride | 198 | |
| F | 3,4-Benzocoumarin | 196 | |
| G | 2,4-Dihydroxybenzoic acid | 154 | |

## 3. Materials and Methods

### 3.1. Chemical and Reagents

Phenanthrene (PHE, $C_{14}H_{10}$, 97.0%, CAS: 85-01-8) was bought from the Energy Chemical Reagent Co., Ltd. (Shanghai, China). Acenaphthylene (ACP, $C_{12}H_{10}$, 98.0%, CAS: 208-96-8) was obtained from Shanghai Acmec Biochemical Co., Ltd. (Shanghai, China). Methanol (MeOH, ≥99.5%, CAS: 67-56-1) and acetonitrile (ACN, ≥99.9%, CAS: 75-05-8) were purchased from Tianjin ZhiYuan Reagent Co., Ltd. (Tianjin, China). Trichloromethane (CF, ≥99.0%, CAS: 67-66-3) was purchased from Chengdu Colon Co., Ltd. (Chengdu, China). Pyrene (PYR, $C_{16}H_{10}$, 97.0%, CAS: 129-00-0) and alcohol (TBA, ≥99.0%, CAS: 442663-47-0) were obtained from the Macklin Biochemical Technology Co., Ltd. (Shanghai,

China). Sodium persulfate ($Na_2S_2O_8$, 99.0%, CAS: 7775-27-1), furfuryl alcohol (FFA, 98%, CAS: 98-00-0), anthracene (ANT, $C_{14}H_{10}$, ≥99.0%, CAS: 120-12-7), fluoranthene (FLUA, $C_{16}H_{10}$, 98%, CAS: 206-44-0), fluorene (FLU, $C_{13}H_{10}$, 97.0%, CAS: 86911-17-3), acenaphthene (ACE, $C_{12}H_8$, 98.0%, CAS: 83-32-9), phenol ($C_6H_5OH$, ≥99.0%, CAS: 108-95-2), and peroxymonosulfate (PMS, $KHSO_5 \cdot 0.5KHSO_4 \cdot 0.5K_2SO_4$, 99.0%, CAS: 22047-43-4) were purchased from Aladdin Biochemical Technology Co., Ltd. (Shanghai, China). Commercial nZVI powders were purchased from Flance (Beijing) nanotechnology Co., Ltd. (Beijing, China).

### 3.2. nZVI-CAC Preparation

The crushed coconut shell was screened and dried for 2 h to remove water, then the dried powder was heated to 400 °C at a rate of 10 °C min$^{-1}$ under anoxic conditions, and kept for calcination for 2 h to obtain carbon materials. To activate the carbon materials, the obtained carbon was heated to 400 °C with inert argon gas and then carbonized for 2 h with the argon gas switched to $CO_2$ and air. The activated material obtained was washed with deionized water to neutralize and dried at 110 °C for 4 h to obtain activated carbon. The activated carbon was placed in a quartz tube, heated to 700 °C at a rate of 8 °C min$^{-1}$ in inert gas, annealed for 2 h to obtain carbonylated activated carbon, and named CAC.

The CAC pellets were firstly ground and passed through a 60-mesh sieve to obtain CAC powder, after which 0.756 g CAC powder was immersed in 100 mL $FeSO_4 \cdot 7H_2O$ solution (0.135 M) and stirred thoroughly for 30 min in a three-neck burner under $N_2$ atmosphere, and then 100 mL $NaBH_4$ (0.27 M) was added in a droplet. After the reaction, the solids were washed three times with anhydrous ethanol and dried under vacuum at 60 °C for 10 h to obtain nZVI-CAC composites with a mass ratio of Fe/CAC of 1:1. To obtain materials with different mass ratios of Fe/CAC, the CAC powder was fixed at 1 g and the added $FeSO_4 \cdot 7H_2O$ was adjusted to a predetermined concentration according to the mass ratio of Fe/CAC.

### 3.3. Soil Spiking

The uncontaminated soil with a depth of 10−30 cm was collected from the East campus of Yangtze University. The physical–chemical characteristics of the soil are listed in Table 3. PHE-contaminated soil was obtained by mixing uncontaminated soil with PHE in methanol solution (1/1, *w/v*), placing it in a 200-rpm shaker for 1 day, and air-drying until the solvent was completely volatile. The preparation procedure of the other soils contaminated with other PAHs was the same as that of the PHE-contaminated soil, except for ANT and PYR, which were dissolved by ACN. All contaminated soil samples used in this study were aged for one week before use. To determine the concentration of PAHs in the spiked soil, the contaminated soil was thoroughly mixed with acetone and sonicated for 20 min. After centrifugation at a speed of 4000 r min$^{-1}$ for 10 min, the supernatant was removed for the determination of the dosage of PAHs, the residual solid was extracted for the second time by acetone. The extraction was conducted three times to completely extract the PAHs from the soil. The concentration of PAHs in the filtrate was determined using a Jimadzu high-performance liquid chromatography system (HPLC). The concentrations of PHE, ANT, FLUA, FLU, PYR, ACP, and ACE in the spiked soil were 94.2 ± 3.2 mg kg$^{-1}$, 93.2 ± 2.3 mg kg$^{-1}$, 92.1 ± 2.7 mg kg$^{-1}$, 84.5 ± 3.8 mg kg$^{-1}$, 98.7 ± 1.2 mg kg$^{-1}$, 87.9 ± 2.1 mg kg$^{-1}$, and 86.2 ± 2.6 mg kg$^{-1}$.

**Table 3.** Primary physical and chemical properties of the collected soil.

| Main Properties | Value |
|---|---|
| pH [a] | 8.02 |
| Organic carbon (%) [b] | 3.65 |
| Organic carbon from soils contaminated with PHE (%) [b] | 3.67 |
| Cation exchange capacity (cmol kg$^{-1}$) [c] | 0.98 |
| Particle size distribution (%) [d] | |
| Silt (2~62 μm) | 76.90 |
| Sand (>63 μm) | 23.10 |
| Heavy metals (mg kg$^{-1}$) [e] | |
| Cr (total) | 26.51 ± 1.45 |
| Cu | 21.24 |
| Pb | 46.23 ± 2.36 |
| Cd | <LOD [g] |
| Contaminants (mg kg$^{-1}$) [f] | |
| PAHs | <LOD [g] |

[a] Water to soil ratio = 5 mL g$^{-1}$. [b] Potassium dichromate-sulfuric acid digestion. [c] Hexamminecobalt trichloride solution. [d] Microtrac S3500 apparatus. [e] Digestion by HCl, HF, and HClO$_4$, monitored by an atomic absorption spectrophotometer AA-7003 (Ewai-group, Beijing, China). [f] Extraction by acetone, measured by an HPLC system. [g] Limit of detection.

### 3.4. Batch Experiments

The experiments were carried out in batches in 100 mL glass bottles. To ensure the homogeneous mixing of the soil slurry, an aqueous–soil mixture containing 2.0 g of contaminated soil and 8 mL of deionized water was oscillated on a thermostatic shaker at a fixed speed. The reaction was timed when the oxidant and catalyst were added to the slurry, and the experiments were protected from light throughout. To determine the degradation of PAHs, a multiple extraction method was used for the extraction of residual PAHs after the reaction [95–98]. At the end of the reaction, excess MeOH or ACN (20 mL) was added, and the bottles were sealed for shaking for 2 h. After sonication for 20 min, the mixture was centrifuged, and the concentration of PAHs in the separated filtrate was determined. The excess supernatant was discarded and the extraction was repeated twice with the residual solid to completely extract PAHs from the soil sample [95–98]. The residual concentration of PAHs in the soil was calculated from the sum of the PAH concentrations of the filtrates.

### 3.5. Analytic Methods and Characterization

The PAHs were quantified by an HPLC system equipped with an Inertsil ODS-SP C18 column (150 × 4.6 mm, 5 μm) with a mobile phase of a mixture of 80.0% acetonitrile and 20.0% ultrapure water, and the total flow rate was 1.2 mL min$^{-1}$.

Scanning electron microscopy (SEM, TESCAN MIRA LMS, Brno, Czech Republic) and transmission electron microscopy (TEM, FEI Talos F200S, Hillsboro, OR, USA) were used to observe the morphology characteristics of the samples, and Fourier transform infrared spectroscopy (FTIR, Nicolet 6700, Thermo Fisher Scientific, Waltham, MA, USA) was employed to characterize the types of functional groups on the surface of the samples. X-ray diffractometer (XRD, Panalytical Empyrean, Nottingham, UK) was conducted to characterize the crystal type of the sample, while X-ray photoelectron spectroscopy (XPS, K-Alpha, Thermo Scientific, Waltham, MA, USA) was used to analyze the composition of the prepared materials. The specific surface areas of the catalysts were analyzed by Brunauer–Emmett–Teller (BET, Micromeritics APSP 2460, Micromeritics, Norcross, GA, USA).

The AGREE software was used to evaluate the greenness of the analysis method [61].

*3.6. Calculation Method*

The removal efficiency of PAHs was described by $(C_0 - C)/C_0$, where $C_0$ denoted the initial pollutant, using the following equation,

$$\text{Removal} = \frac{C_0 - C}{C_0} \times 100\% \tag{9}$$

where $C_0$ and $C$ represent the concentration of PAHs in the soil before and after the reaction in mg kg$^{-1}$, respectively.

The pseudo-first-order kinetic model was used to describe the degradation rate of PHE, using the following equation [96]:

$$-\ln\left(\frac{C_t}{C_0}\right) = kt \tag{10}$$

where k is the reaction rate constant, min$^{-1}$; $C_t$ is the concentration of PHE in the soil at time t min, mg kg$^{-1}$.

**4. Conclusions**

In this study, for the first time, an nZVI-CAC composite was synthesized and employed for the remediation of PAH-contaminated soil. The characterizations of XRD, FTIR, SEM, and XPS for the prepared materials demonstrated the successful loading of Fe$^0$ and the composite nZVI-CAC was rich in oxygen-containing functional groups, including C=O. The PS/nZVI-CAC system was found to be superior for PHE oxidation over the processes using commercial nZVI or those with oxidants of $H_2O_2$ and PMS. At the optimal conditions, with an Fe/CAC ratio of 1:1, a PS concertation of 100 mmol kg$^{-1}$, and an nZVI-CAC dosage of 10 g kg$^{-1}$, seven tested PAHs with 3–4 rings were effectively degraded, and the removal efficiencies were in the range of 60.8–90.7%. Finally, based on the scavenging experiments, surface-bound $SO_4^{-\bullet}$ was verified to be more contributive than $HO^\bullet$ to PHE oxidation, while $O_2^{-\bullet}$ partially improved PHE degradation and $^1O_2$ played an insignificant role in PHE remediation. Consequently, the findings of this study enable a cost-effective approach for the remediation of PAH-polluted soil by ISCO with nZVI-CAC. Moreover, when using the nZVI-CAC for actual site remediation, nZVI could be gradually oxidized to iron oxide after being injected into the restoration area [99], and the carbon materials can improve soil fertility and increase crop yields [100,101]. Therefore, the residue of nZVI-CAC in site application is expected to have less impact on soil systems.

**Author Contributions:** Conceptualization, C.C., Y.T. and E.B.; data curation: C.C., Z.Y., Y.T., S.S., J.X. and K.Z.; resources: Y.Z., R.Z. and E.B.; writing and original draft preparation: C.C. and Z.Y.; writing review and editing: Y.T. and Q.S. All authors have read and agreed to the published version of the manuscript.

**Funding:** This study was supported by the National Key R&D Program of China (Grant No. 2023YFC3708002), the scientific research foundation from the Department of Education of Hubei Province (Grant No. Q20211311), and the Science and Technology Innovation Fund from CNPC (2022DJ6906).

**Data Availability Statement:** The datasets used and analyzed in this study are available from the corresponding author upon reasonable request.

**Conflicts of Interest:** The authors have no relevant financial or non-financial interests to disclose.

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
