# Peer review of "Remediation of Polycyclic Aromatic Hydrocarbon-Contaminated Soil by Using Activated Persulfate with Carbonylated Activated Carbon Supported Nanoscale Zero-Valent Iron"

_catalysts, doi:10.3390/catal14050311_

Round 1

Reviewer 1 Report

Comments and Suggestions for Authors

The work is devoted to the solution of an important modern problem associated with the purification of soils from organic polycyclic pollutants. The authors have applied an interesting approach to solve the set problems, associated with the use of composite materials, interest in which is due to the unique properties of the composites under consideration. The article is well written, but requires a number of additions and clarifications:

1.- The authors did not say anything directly about the toxic properties of persulphate (PS) and its derivatives.

2.- Reflexes on XRD are very wide, also the determination of the phase composition on a reflex is not quite correct. It is necessary to confirm by alternative methods.

3.- The sentence (line 175-176) about the interaction mechanism should be clarified.

4.- The introduction/conclusion should state the authors' position on direct land clearing. How will the composite material be removed from the land after clearing?

5.- The abbreviation ROS is not deciphered.

6.- Reflections on value-based choices are better placed in the introduction.

7.- What is Removal(%) and how is it calculated? A calculation formula must be added.

8.- It is necessary to explain how the dose of nZVI-CAC was chosen.

9.- What are the decomposition products of PAH.

10.- The abbreviations TBA, FFA should be deciphered beforehand.

11.- Why PHE was chosen for the experiments. Need to explain.

12.- What is meant by "Selectivity". What are the processes involved. How it is calculated.

13.- In the experimental part, the countries of manufacture should be added to the instrument companies.

Comments on the Quality of English Language

Author Response

请参阅附件

Reviewer 2 Report

Comments and Suggestions for Authors

Dear Editor,

The manuscript (catalysts-2907218) entitled ‘‘Remediation of polycyclic aromatic hydrocarbons contaminated soil by using activated persulfate with carbonylated activated carbon supported nanoscale zero-valent iron’’ can be published in the Catalysts after moderate revision.     My suggestions:

1-     Line 111-113,  The distinct absorption peaks at 3430 cm−1, 3130 cm−1 and 1400 cm−1 for both samples represented the -OH stretching vibration, -CH2 asymmetric stretching vibration, and C=O stretching vibration, respectively [31]. Some peak comments are not true. The carbonyl peak gives asymmetric and symmetric vibrational peaks. So it cannot be symmetrical without asymmetrical stretching. The asymmetric one gives higher peak intensity and comes in the 1600-1750 cm-1 region depending on the chemical environment and molecular weight of the molecule to which it is attached. The symmetric stretching peak comes at 1350-1450 cm-1 with lower intensity. As far as I understand, the small peak at 1630 cm-1 belongs to asymmetric C=O and the peak at 1400 cm-1 belongs to symmetric CO, C-H bending peak, an OH in plane bend. Thus, the symmetrical CO peak became more intense because the C-H bending peak and OH in plane bend overlapped. If this is not the case, there is no carbonyl group in your compound. Do not forget to make the corrections in Figure 1b. I suggest one reference for FTIR interpretation. Journal of Inorganic and Organometallic Polymers and Materials31, 3613-3623, 2021.

2-     In line 284, Correct that there is no charge equivalence in Equation 6. The left side has 2 negative charges, the right side has 3 negative charges.

 nZVI-CAC-C=O + S2O82− → nZVI-CAC-C-O* + SO4 −• + SO42−

3-     In lines 128-129, As shown in Figure 2(c−d), the N2 isotherms of CAC displayed a typical microporous carbon material (IUPAC, type I). I don’t think it is type I. I think it looks like type IV, because there is hysteresis loop and it limits uptake of high P/Po. It could be mesoporous.

Reviewer 3 Report

Comments and Suggestions for Authors

1-      Characterization of the materials is not completed. TEM and EDS analysis should be added.

2-      In article emphasis on remediation (degradation or adsorption) of the analyst(s) but there is no prove for that. How the authors differentiated adsorption and degradation? There should be spectra before and after the process.

3-      COD analysis is necessary to clarify the mechanism.

4-      FT-IR of the materials before and after the process is necessary.

5-      What is the advantageous of this method in comparison with others? A section of comparison with other methods should be added.

6-      What is the kinetic of degradation? For better understanding the process, it is my suggestion to add that.

7-      The concept of the article is environmental; accordingly, Analytical Greenness Metric can be interesting for future reader. Following article may be helpful

https://pubs.rsc.org/en/content/articlelanding/2024/ay/d3ay02116e/unauth

8-      There are some minor grammatical mistakes. The article should be checked carefully.

Comments on the Quality of English Language

There are minor grammatical mistakes.

Round 2

Reviewer 2 Report

Comments and Suggestions for Authors

Dear Editor,

The manuscript (catalysts-2907218R1) entitled ‘‘Remediation of polycyclic aromatic hydrocarbons contaminated soil by using activated persulfate with carbonylated activated carbon supported nanoscale zero-valent iron’’ can be published in the Catalysts after minor revision. Authors made necessary corrections step by step. However, a few minor errors were introduced during editing, which should have been corrected before the article was published. My suggestions: 1- line 57, efficiency[13–15].   efficiency[13-15] is underlined, remove the underline. 2- Figure 8 is numbered twice. The second Figure 8 should be numbered as Figure 10. Line 355, ……oxidative remediation of organic contaminants was proposed in Figure 8. Change Figure 8 to Figure 10. Line 365, Figure 8. Mechanism of PS/nZVI-CAC system for PHE oxidation. So there should be a correction in the Figure caption, as Figure 10 (not Figure 8). 3- Lines 370-385, Why do the names of chemical compounds such as 9-Phenanthrol, 2,2'-Biphenyldicarboxylic anhydride, and Biphenyldicarboxylic acid start with a capital letter in a sentence?

Please capitalize at the beginning of the sentence and lowercase in the sentence.

4- Line 481, where k is the reaction rate constant, h-1 ; Ct is the concentration of PHE in the soil at time t min, mgkg-1. The reaction rate constant,  h-1, can it be min-1 instead of h-1? Please check.

Author Response

请参阅附件。
